# Keratinocytes: An Enigmatic Factor in Atopic Dermatitis

**DOI:** 10.3390/cells11101683

**Published:** 2022-05-19

**Authors:** Pamelika Das, Pappula Mounika, Manoj Limbraj Yellurkar, Vani Sai Prasanna, Sulogna Sarkar, Ravichandiran Velayutham, Somasundaram Arumugam

**Affiliations:** Department of Pharmacology and Toxicology, National Institute of Pharmaceutical Education and Research (NIPER)-Kolkata, Chunilal Bhawan, 168 Maniktala Main Road, Kolkata 700054, West Bengal, India; pamelikadas@gmail.com (P.D.); mounika27.pappula@gmail.com (P.M.); manoj.yellurkar86@gmail.com (M.L.Y.); vsprasanna92@gmail.com (V.S.P.); shulognas44@gmail.com (S.S.)

**Keywords:** keratinocytes, skin barrier, atopic dermatitis, immune dysregulation, therapeutic target

## Abstract

Atopic dermatitis (AD), characterized by rashes, itching, and pruritus, is a chronic inflammatory condition of the skin with a marked infiltration of inflammatory cells into the lesion. It usually commences in early childhood and coexists with other atopic diseases such as allergic rhinitis, bronchial asthma, allergic conjunctivitis, etc. With a prevalence rate of 1–20% in adults and children worldwide, AD is gradually becoming a major health concern. Immunological aspects have been frequently focused on in the pathogenesis of AD, including the role of the epidermal barrier and the consequent abnormal cytokine expressions. Disrupted epidermal barriers, as well as allergic triggers (food allergy), contact allergens, irritants, microbes, aggravating factors, and ultraviolet light directly initiate the inflammatory response by inducing epidermal keratinocytes, resulting in the abnormal release of various pro-inflammatory mediators, inflammatory cytokines, and chemokines from keratinocytes. In addition, abnormal proteinases, gene mutations, or single nucleotide polymorphisms (SNP) affecting the function of the epidermal barrier can also contribute towards disease pathophysiology. Apart from this, imbalances in cholinergic or adrenergic responses in the epidermis or the role played by immune cells in the epidermis such as Langerhans cells or antigen-presenting cells can also aggravate pathophysiology. The dearth of specific biomarkers for proper diagnosis and the lack of a permanent cure for AD necessitate investigation in this area. In this context, the widespread role played by keratinocytes in the pathogenesis of AD will be reviewed in this article to facilitate the opening up of new avenues of treatment for AD.

## 1. Introduction

Keratinocytes are the major cells in the human epidermis and are involved in skin barrier functions. They protect the body against external environmental stimuli, microbes, and pathogens. Pattern recognition receptors on the surface of keratinocytes aid in the recognition of pathogens’ entry into the skin and are responsible for immune responses [1]. The expression of the keratinocytes can be increased by various allergens, microorganisms, wounds, and some triggering factors [2]. The key cytokines regulating the development of inflammatory responses in keratinocytes are tumor necrosis factor (TNF)-α and interferon (IFN)-γ. Several antimicrobial peptides produced by keratinocytes, such as cathelicidin and human defensins, play an important role in inflammation and wound healing [3]. Keratinocytes have always been a critical element in the regulation of skin pathology in atopic dermatitis (AD).

Initiating from early infancy, AD represents a chronic inflammatory skin disorder undergoing a number of relapses and recoveries throughout the time period of infection and is particularly characterized by eczematous lesions with intense itching. AD is becoming more common, with a prevalence rate in children ranging from 0.92–26.6% and in adults ranging from 1.21–17.1% [2]. It can be triggered by various factors such as allergens, irritative substances, and infectious microbes such as *Malassezia* species and *Staphylococcus aureus*, causing epidermal barrier alterations and dysregulation of helper T cell (Th2) immune response. Infiltration of the inflammatory cells, thickening of the epidermis, and upregulated levels of Th2 cytokines are prominent in AD.

Keratinocytes are considered the key effector cells in AD, triggering abnormal immune responses [4,5]. They produce inflammatory cytokines such as interleukin (IL)-1β, IL-6, IL-8, IL-18, and IL-25 and chemokines such as thymus and activation-regulated chemokine (TARC), cutaneous T cell attracting chemokine (CTACK), and monocyte derived chemokine (MDC) in skin inflammation [6,7,8]. Elevated levels of these inflammatory mediators and infiltration or accumulation of lymphocytes, macrophages, mast cells, eosinophils, and dendritic cells in the epidermis is characterized by the progression of inflammatory skin diseases such as AD [9,10,11]. In AD, Th2 cytokines inhibit the expression of antimicrobial peptides. Thymic stromal lymphopoietin (TSLP) is also released by the keratinocytes, and it is highly expressed in AD, responsible for irritation and itching prominent in the disease pathology. TSLP dysregulates Th2 immune responses, increasing the production of inflammatory cytokines like IL-4, IL-5, IL-13, and IL-31 by activating the dendritic cells, thus initiating the inflammatory responses in AD. These are the primary cells playing a major role in the pathogenesis of AD because the cytokines released from the Th2 cells are mainly responsible for increased IgE production, leukocyte migration, activation of mast cells, and eosinophils. The T cell-derived lymphokines trigger the keratinocytes to activate the T cells and leukocytes in the dermis, leading to various inflammatory reactions [8,12]. Recent studies have revealed that acetylcholine levels are also elevated in AD. Various cytokines are responsible for the production of acetylcholine in the epidermis [13]. Filaggrin (FLG), a protein involved in the formation and maintenance of the epidermal barrier, also plays an important role in the pathophysiology of AD. Any alteration in FLG may lead to aggregation of keratin filaments and alteration in the skin pH values, dehydrating the skin and thereby inducing xerosis and pruritus [14]. On the other hand, keratinocytes produce cytokines and immunosuppressive factors that are able to block the inflammatory responses mediated by keratinocytes themselves [15]. Another important inflammatory mediator produced by keratinocytes is granulocyte macrophage colony-stimulating factor (GM-CSF), which is overexpressed in AD and is regulated by other cytokines such as IFN-γ, TNF-α and IL-17 [16].

Keeping in view these inexplicable effects of keratinocytes in AD, this review will provide a detailed perspective on the critical role of keratinocytes in AD, both as a first-line defense neutralizing the initial pathological insults and also as a mediator aggravating the disease pathogenesis.

## 2. Factors Triggering Keratinocyte Activation

There are usually two alternative pathways for epidermal keratinocytes: differentiation and activation. Keratinocyte differentiation is usually dominant in the healthy epidermis. Keratinocytes are not activated in the healthy epidermis; they proliferate and differentiate in the basal and suprabasal layers of skin. Certain pathological or inflammatory conditions lead to the migration and hyperproliferation of keratinocytes at the site of inflammation. Cytokines and growth factors such as IL-1β, TNF-α, IFN-γ, transforming growth factors (TGF-α and TGF-β), GM-CSF, as well as ultraviolet-B (UVB) radiation, allergens, haptens, lipopolysaccharides, lysophosphatidic acid, and others can stimulate keratinocyte activation [8,17]. Activation leads to an alteration in the cytoskeleton of keratinocytes, making them hyperproliferative and upregulating the expression of cell surface receptors. The activated cells send paracrine signals to lymphocytes, fibroblasts, melanocytes, and endothelial cells, which manifest a synchronized repair response, causing re-epithelialization at the site of injury, and thereafter revert back to the normal differentiation behavior. This balance between the differentiation and activation of keratinocytes plays a major role in disease pathogenesis and is regulated by various factors.

### 2.1. Cytokines

Inflammation brings about a surge in the cytokine levels in the epidermis. The major cytokines that act as initiators for the activation of keratinocytes are IL-1 and IL-4. IL-1 activated keratinocytes express K6 and K17 keratin proteins, which are released only in the inflammatory epidermis, leading to inflammatory signals distinct from those in the healthy epidermis. After activation, keratinocytes upregulate the synthesis of signaling molecules including cytokines such as TNF-α, TGF-α, GM-CSF, IL-6, and IL-8, facilitating the transfer of signals to surrounding cells in cell injury. These signaling molecules act as paracrine for lymphocytes and autocrine for keratinocytes themselves [16,17,18,19]. IL-17 can also activate epidermal keratinocytes on its own or in combination with TNF-α or IFN-γ. IL-4, another initiator of keratinocyte activation, causes the expression of chemokines such as C-X-C motif chemokine receptor 3 (CXCR3) in T-cells [20]. Cell surface markers including integrins, intracellular adhesion molecule (ICAM)-1, and extracellular matrix components such as fibronectin are also produced by activated keratinocytes, inducing the migration of activated cells. IFN-γ is the most potent cytokine that activates the keratinocytes. In inflammatory skin diseases such as AD, IFN-activated human keratinocytes induce the expression of major histocompatibility complex (MHC) class II molecules and ICAM-1, as well as the synthesis of other class II peptide complexes [8,17]. ICAM-1 is a cell surface glycoprotein responsible for leukocyte adhesion, T cell activation, and recruitment of leukocytes to the site of inflammation in the skin, initiating the inflammatory responses. It is an adhesion receptor and a ligand for the leukocyte function associated antigen 1 (LFA-1) receptor present on leukocytes. ICAM-1 expression is also stimulated by inflammatory cytokines such as IL-4, IL-6, IL-17, IL-18, and TNF-α in response to IFN-γ. Under inflammatory conditions, ICAM-1 is highly expressed by epithelial cells, endothelial cells, and immune cells [8,21]. ICAM-1 expressed by the dendritic cells and macrophages is important for T cell activation. Regeneration of reactive oxygen species (ROS) and activation of phagocytosis are induced by ICAM-1 [21,22,23,24]. Activated keratinocytes also serve as a source for the secretion of chemokines such as CTACK and CXCL8 during inflammation. IFN-γ promotes the expression of chemokines (CXCL8 and CTACK) in conjunction with TNF-α. Keratinocytes also release other inflammatory cytokines that are capable of inducing skin inflammation, such as IL-12 and IL-23 [25]. Primarily, IL-1β is responsible for the increased expression of chemokine C-C motif ligand (CCL) 27 in keratinocytes. During inflammation, activated keratinocytes and macrophage inflammatory protein (a potent chemokine) induce the recruitment of Langerhans cell precursors into the epithelium [26].

IL-22 is also one of the strongest initiators for the activation of keratinocytes in the epidermis. IL-17, IL-12, and IL-22 are the key cytokines involved in the regulation of epidermal functions and are also responsible for the upregulation of expression of genes such as β-defensin, which is directly associated with inflammation [25].

### 2.2. Immunologic Triggers—Allergens

Disruption of epithelial barrier function causes mutations in the filaggrin gene, allowing microbes and allergens to penetrate into the epithelium and initiate inflammatory responses, thereby activating the keratinocytes and causing the release of pro-inflammatory cytokines [27]. Allergen-induced direct activation of keratinocytes is the primary factor, important for the release of inflammatory mediators in the skin cells compared to cytokines released from the infiltrating T cells [8]. Most common allergens include contact allergens, food allergens, inhaled allergens, dyes, weeds, plants, balsams, animal danders, and molds [8,28].

#### 2.2.1. Contact Allergens

Contact allergens mostly include di and trinitro chlorobenzene, ovalbumin, nickel, urushiol, fragrances, and cosmetic products. Exposure to the allergens induces inflammation by increasing the expression of inflammatory mediators such as chemokines, cytokines, and intracellular adhesion molecules, causing direct activation of keratinocytes in the skin, leading to inflammation. Haptens and other allergens induce allergic responses, thereby releasing IL-1, TNF-α and GM-CSF on activation of keratinocytes [8,28]. Allergens lead to sensitization of the skin; that induces the generation of IL-17 producing T cells and elevated serum IL-17 levels. House dust mites, detergents, soap, and some other chemicals cause strong Th2 immune responses, causing cutaneous inflammation. In AD, keratinocytes sense the entry of allergens and initiate the expression of inflammatory mediators in the epidermis [8,27,29]. The ability of barrier function varies with age. Aging diminishes the capability of the epidermis to clear out debris, leading to the accretion of allergens in the epidermis. In acute AD, in response to any contact allergen such as nickel or other antigen, keratinocytes are activated, expressing monocyte chemoattractant protein-1 (MCP-1), high levels of CCL27/CTACK and low levels of chemokines, instigating the accumulation of neutrophils in the inflamed skin [8]. 

#### 2.2.2. Food Allergens

Recent studies have shown that 40% of children develop skin rashes in response to food allergens, inducing moderate to severe AD. Common food allergens from the diet include hen’s eggs, cow’s milk, soya, wheat, fish, peanuts, and cooked potatoes, which encourage the development of skin lesions through the activation of keratinocytes and thereby increase the flares of AD in infants. Allergen-activated keratinocytes are responsible for the production of IgE-mediated allergic reactions through the activation of mast cells. The skin prick test, atopic patch test, and measurement of serum levels of food-specific IgE are used to identify specific allergens associated or correlated with AD. Food allergens increase serum IgE levels before the first year of life, causing skin inflammation [28,30,31]. Children are more prone to allergic reactions due to the increasing variety of foods they consume. Several studies have found that milk, eggs, seafoods, and wheat are major contributors to the development of 90% of food allergies in infants. Ingestion of food allergens produces cutaneous eczematous lesions in AD. Due to a pathophysiological mechanism termed “late eczematous reaction”, lesions are formed after 6 h or after 1 day, in food allergen-induced eczema. It elevates food allergen-specific T cells and CD4+ T cells in response to certain foods [31,32].

#### 2.2.3. Inhalant Allergens

Inhalants often play a prominent role in developing AD by manipulating the keratinocytes. Inhaling animal or horse danders, human danders, cereal flour, house dust mites, silk protein, wool, cooking odors, pollen (ragweed pollen, timothy pollen), and molds activates keratinocytes, leading to increased expression of allergen-specific Th2 cells and decreased Th1 cell responses, exacerbating the release of inflammatory mediators in AD. The role of inhalant allergens as a cause of AD has always been less studied compared to food allergens with respect to the pathophysiology of AD [33]. Children around the age of 7 years are often allergic to inhalants, and these inhalant allergens increase the Th2 cytokines, which in turn are responsible for increasing the IgE levels predominantly. At the age of 2 years or more, studies show that inhaled allergens cause IgE-mediated allergic responses [32,33,34]. Other secondary allergens, such as tobacco, goat hair, orris root, and some insecticides, are occasionally responsible for the development of AD [33,35]. House dust mites (*Dermatophagoides pteronyssinus*) are the major inhaled allergens. Sensitization to these inhalant allergens is found to gradually increase with age [31,36].

## 3. Possible Role of Keratinocytes in AD

Keratinocytes are the prime cell type in the epidermis. The epidermis is composed of 95% keratinocytes and is destined for the maintenance of barrier function, protecting the skin from external injury by the production of keratins and antimicrobial peptides that invade the microbes or pathogens [1,37]. So, it has an important role in skin repair and immune functions. Keratinocytes actively contribute to the damage repair of skin via the release of certain proteins such as pro-inflammatory cytokines and chemokines. These are responsible for the recruitment of neutrophils and macrophages to the site of damage or injury, aimed at removing cellular debris and undergoing phagocytosis to destroy the pathogens or microbes. The re-epithelialization process includes migration, proliferation, and remodeling of keratinocytes, leading to restoration of the damaged epidermal barrier [38,39,40]. Keratinocytes also produce a number of antimicrobial peptides that have the ability to kill pathogenic bacteria and fungi. Recent studies revealed that toll-like receptors (TLRs) are mostly expressed by keratinocytes, such as TLR-1, TLR-2, TLR-3, TLR-4, TLR-5, TLR-6, and TLR-9, retinoic-acid-inducible gene-I-like receptors, and C-type lectin receptors, and play an important role in the initiation of antimicrobial effects mediated by keratinocytes. In inflammatory conditions, the nuclear factor kappa B (NFκB) pathway is also activated by TLR signaling, resulting in the release of cytokines, chemokines, and antimicrobial peptides [38,41,42]. Dysregulation of this immune response converts the first line of defense into an active contributor to disease pathogenesis in AD.

## 4. Keratinocytes as a Guardian of Skin Immune Defense

The epidermis, a semi-permeable membrane consisting of keratinized squamous epithelial cells of the skin, is what actually comprises the skin barrier. This keratinized epithelial layer guards against the constant microbial, allergic, mechanical, and chemical insults that our skin has to endure day by day. The two important key factors that contribute to barrier function are the diffusion of critical molecules across the semi-permeable membrane and the prevention of trans-epidermal water loss (TEWL). Loss of this barrier function is the key factor behind several skin inflammatory disorders like AD.

Microbiome: The cutaneous microbiome is one of the major contributors to the skin barrier. A number of commensal bacteria, including *Burkholderia* spp. and *Lactobacilli*, *Staphylococcus aureus*, and *Staphylococcus hominis* are part of this microbiome. The mechanism by which these microbiota interact with the barrier sites of the epithelium mostly involves a ligand-gated transcription factor called aryl-hydrocarbon receptor (AHR), abundant in epidermal keratinocytes. These receptors can sense the environment and induce the metabolism of xenobiotics by expressing an AHR gene battery involving members of the cytochrome p450 family of enzymes such as CYP1A1 (Figure 1). AHR is considered to be a prominent mechanism of interaction between the host and microbiota, ensuring homeostasis. Moreover, activation of AHR by physiological metabolites derived from tryptophan has been proven to be beneficial in preclinical studies of AD [43]. It also initiates repair of the skin barrier, as observed in the case of coal tar, activating this pathway and becoming one of the most prominent treatment strategies for AD [44]. Various antimicrobial peptides are also synthesized by keratinocytes, including human beta defensins (HBD-3) and cathelicidin (LL-37), which demonstrate their antibacterial activity functioning through multiple mechanisms [45]. The expression of these anti-microbial peptides (AMPs) is upregulated in the case of any secondary infection or inflammation, forming an innate chemical barrier against pathogens [46]. HBD-3 improves the tight junction barrier by expressing several claudins and, in addition to LL-37, it induces the expression of macrophage inflammatory protein-3 alpha, IL-6, IL-18, IL-10, and regulated upon activation, normal T cell expressed and secreted (RANTES) [47,48]. They also stimulate mitogen-activated protein kinase, facilitating keratinocyte proliferation, migration, re-epithelialization, and angiogenesis, thereby preventing inflammation, causing wound healing, and regulating the skin barrier [49] (Figure 1).

Pattern recognition receptors (PRR): These receptors are critical for the detection of a wide range of pathogen-associated molecular patterns (PAMPs) such as β-glucans, lipopolysaccharides (LPS), lipopeptides, nucleic acids of various viruses, fungi, and Gram-positive or Gram-negative bacteria [50]. There are various types of PRRs expressed by keratinocytes, including nucleotide-binding oligomerization domain-like receptors (NLRs), TLRs, C-type lectin receptors (CLRs) and RIG-I-like receptors (RLRs) [51] (Figure 1).

The NLR family of PRRs are intracellular sensors of cellular damage and bacterial infection, including nucleotide-binding oligomerization domain, leucine rich repeat and pyrin domain containing (NLRP), NLRC, and NOD-containing families [52]. On activation, the NLRP3 inflammasome expressed in keratinocytes stimulates inflammatory caspases, resulting in the release of IL-18 and IL-1β [53]. House dust mites, UVB-induced DNA damage, and pesticides have all been shown to be potent activators of NLRP3 in keratinocytes [54,55]. NLRP1, a member of the NOD family, has been recognized to be a predominant part of keratinocytes, playing a critical role in sensing UV-B and the secretion of subsequent cytokines by human keratinocytes [52]. Two functional members of the NOD-containing family, NOD1 and NOD2, present in human keratinocytes, can sense bacterial peptidoglycan fragments such as muramyl dipeptide and γ-glutamyl-diaminopimelic acid, inducing the secretion of CXCL8.

TLRs, being the most abundant type of PRRs expressed by activated keratinocytes, are able to detect various kinds of pathogens, such as TLR5 recognizing bacterial flagellin and TLR4 detecting LPS-induced translocation of NF-κB and secretion of IL-8 [56,57]. TLR2 can also recognize bacterial components and initiate the production of proinflammatory cytokines [56]. With the exception of TLR3, ligand recognition by TLRs leads to induction of downstream signaling cascades including the MYD88 complex by keratinocytes, thereby initiating immune responses by triggering the production of various cytokines, chemokines, and AMP [42]. TLRs can also recognize viral or bacterial nucleic acids broken up and incorporated inside the cell, especially the intracellular TLRs such as TLR9, detecting CpG-methylated DNA and inducing the production of type I IFNs, CXCL10, and CXCL9 selectively [56]. TLR3 has the ability to bind to viral double-stranded RNA and initiate an immune response [58].

Dectin-1, or C-type lectin domain family 7 member A, is a type of PRR expressed in human keratinocytes that can detect fungal antigens such as *Mycobacterium ulcerans* and β-glucan in a TLR2 dependent mechanism [59]. On binding with β-glucan, CLRs enhance differentiation, proliferation, and migration of keratinocytes, which points out that dectin-1 can be a potential therapeutic target in wound healing [60].

RLRs are critical for host anti-viral defense, sensing viral antigens and inducing type I interferon production. A cytosolic PRR, RIG-1, is crucial for this type of anti-viral response. Keratinocytes are capable of sensing double-stranded viral RNA, thereby expressing RIG-1 and melanoma differentiation-associated protein 5 (MDA-5), activating type I interferon responses [61]. Apart from RIG-1, other RLRs include LGP2 and MDA-5 [62]. On exposure to IFN-γ, keratinocytes undergo an alteration in their phenotype, expressing MHC-II molecules on their surface. They behave as antigen-presenting cells, inducing functional responses for CD4+ and CD8+ memory T cells [63]. The inherent MHC-II expression on keratinocytes can regulate Th1 responses induced by commensals but does not have any effect on T helper 17 (Th17) responses [64]. This reflects the sovereign role of keratinocytes in host–microbiota interactions. Furthermore, a guanine nucleotide exchange factor for Rho GTPases named dedicator of cytokinesis 8 (DOCK8) can also monitor the aforesaid host–microbiota interaction and immune cell trafficking [65]. Hence, a deficiency of this factor leads to a significant increase in cutaneous viral load, atopy, and marked infection [66].

Mechanical stimulation: Apart from microbes, mechanical stimuli such as stretching or pressure also have an effect on the immune response of keratinocytes. Keratinocytes, on treatment with mechanical stimulation, induce the secretion of various inflammatory chemokines (CCL20 and CXCL1) and cytokines (IL-23, IL-1α, IL-6, and TNF) [67]. Studies have also further reported that mechanical stimulation enhances the proliferation of keratinocytes and prevents differentiation [68] (Figure 1).

## 5. Keratinocytes as a Contributor to Pathogenesis in AD

AD is a chronic inflammatory disease, a complex of altered immune responses, infiltration of the inflammatory cells, and genetic factors leading to epidermal barrier dysfunction, with increased sensitization to allergens [69]. Dry skin is caused by a decrease in the expression of epidermal lipids such as ceramides and long-chain fatty acids, as well as epidermal proteins such as filaggrin, keratin, loricrin, involucrin, and claudins. It increases the penetration of microbes and allergens into the epidermis, thereby activating the keratinocytes [70]. Activated keratinocytes manifest a surge in production of chemokines that attract T cells, cytokines, and other inflammatory and proinflammatory mediators pertinent to inducing inflammation in AD. The immune responses produced by the allergens are different from the Th2 immune responses [8,70,71].

Inflammatory cytokines and chemokines play a major role in the pathogenesis of AD. Chemokines recruit leukocytes to the injured area or inflammatory tissue. Activation of keratinocytes by IFN-γ induces the production of chemokines such as CCL5/RANTES, CTACK/CCL27, and CXCL10/IP-1, whose levels are highly upregulated in AD. These are responsible for the infiltration of eosinophils and leukocytes and the release of histamine by the activation of mast cells [8]. The expression of surface molecules such as MHC class II and ICAM 1 is also induced by IFN-γ. Activated keratinocytes significantly increase the release of pro-inflammatory cytokines such as IL-1, IL-25, IL-33, TSLP, and TARC in the epidermis, leading to the activation of Langerhans cells and dendritic cells in the epidermis and dermis. Thereafter, activated dendritic cells stimulate Th2 cells, aggravating the production of inflammatory cytokines such as IL-4, IL-5, IL-13, and IL-31, causing increased keratinocyte differentiation and dysregulation of immune response [8,12]. These cytokine levels are markedly upregulated in inflammation [70,72,73]. In addition, dendritic cells also stimulate Th1, Th17, and Th22 cytokines, resulting in hyperproliferation of keratinocytes and alterations in the expression of the filaggrin gene. IL-4, IL-13, and TSLP additionally facilitate the downregulation of epidermal proteins such as filaggrin, keratins, and intracellular proteins. These cytokines also play an essential role in the reduction of anti-microbial peptides and immune dysregulation [70,74,75,76]. TSLP is a cytokine significantly produced by keratinocytes. TSLP is highly expressed by the epithelial cells and keratinocytes when keratinocytes are exposed to an allergen. TNF-, Th2-cytokines, and other proinflammatory mediators also promote TSLP upregulation in epithelial cells. TSLP in turn induces Th2 immune responses by directly activating dendritic cells. It also stimulates the differentiation and maturation of Th2 cells from T cells and enhances the production of Th2 cytokines (IL-4, IL-5, and IL-13). Dendritic cells activated by TSLP increase sensitization of the skin to allergens and activate primary and secondary immune responses [12,70]. IL-4 and IL-13 induce the synthesis of IgE, whereas IL-5 is responsible for the recruitment of eosinophils [77]. IL-13 also enhances matrix metalloproteinase (MMP)-9 expression in keratinocytes, inducing migration of leukocytes into the epidermis. It downregulates the expression of filaggrin protein [73]. Recent studies have revealed that IL-17 producing CD4+ Th17 cells have a significant role in the pathogenesis of inflammatory diseases such as psoriasis and AD. CD4+ T cells also have a role in the development of skin inflammation by recruiting eosinophils in the lesional skin and inducing the production of Th1 and Th2 cytokines [78]. These cells are also capable of inducing IgE synthesis by B cells, fostering the survival of eosinophils [70,79,80,81,82]. Additionally, dendritic cells also induce the expression of IL-23 and are responsible for the secretion of IL-17 by native T cells. IL-17 induces production of proinflammatory and inflammatory chemokines and cytokines, including IL-6, macrophage inflammatory protein (MIP), MCP, TNF-α, GM-CSF, and antimicrobial peptides by keratinocytes [83]. IL-31 is a new Th2 cytokine, produced by the infiltrating T cells in the skin, and it is highly expressed in AD inflammation. This cytokine expression is induced by histamine and leads to itching in diseased individuals [84]. Studies have demonstrated that disruption of the epidermal barrier upregulates MIP-3α mRNA, a major mechanism for the activation of dendritic cells and T cells in inflammation [5] (Figure 2).

Bacterial and viral infections are also frequent complications found to be associated with AD. The defective skin barrier along with the inability of keratinocytes to combat secondary skin infections are the major reasons behind these complications. There are abnormal bacterial colony formations, especially of Staphylococcus aureus, in the cases of more severe AD patients, and Staphylococcus epidermidis colonization is found in less severe cases [85]. In normal human skin, commensal bacteria try to neutralize these infections by inducing keratinocytes to produce AMPs such as HBD-3 and LL-37, whereas the keratinocytes of diseased individuals are incapable of producing these effector antimicrobial molecules [45]. Hence, cutaneous dysbiosis affects the innate immunity of the skin, causing severe inflammation [85]. In addition, suppression of AMPs by IL-4, IL-13, and Th-2 cytokines aggravates the condition, altering the skin pH and causing chronic microbial infections [86,87]. Several other factors, such as FLG, proteases, microbes, cytokines, and enzymes, also regulate skin pH, which is a major factor in monitoring skin homeostasis [88]. Alkaline pH aids in microbial growth and facilitates skin barrier defects [88]. Furthermore, primary toxins released by the predominant Staphylococcus aureus inhabiting AD patients, named staphylococcal alpha toxin, add to the toxicity, causing necrosis and defects in the skin barrier [89]. Further investigation would shed some light on how dysbiosis becomes a deciding factor in epidermal barrier function. A defective lipid matrix can also regulate the colonization status of Staphylococcus aureus in AD patients [90]. Alteration in the expression of enzymes involved in the biosynthesis of ceramides and free fatty acids in the stratum corneum is responsible for impairing the skin barrier function [91]. Omega-hydroxyceramides, a class of synthetic compound, are reported to stimulate the differentiation of keratinocytes, accelerating the recovery of skin barrier function and improving the integrity of the epidermis [92]. Periodic lipid replacement has also been shown in studies to reduce approximately 50% of AD relapse cases [93]. Hence, the lipid matrix of the stratum corneum can be considered as a prominent contributor to skin barrier function.

## 6. Therapeutic Implications

AD has been basically considered to be a skin disorder for which anti-inflammatory drugs should be the first-line therapy, including various corticosteroids and immunosuppressants such as tacrolimus, with the recent addition of crisaborole, a phosphodiesterase-4 inhibitor. In the case of the more severe forms of AD, contemporary therapeutic guidelines indicate the use of azathioprine, cyclosporine, mycophenolate mofetil, and methotrexate [94]. However, the increased prevalence of AD in children, with its unpredictable progression and the limited availability of safe, approved drugs, is a major constraint in the treatment of AD.

A thorough understanding of the underlying pathological mechanism shed some light on potential drug targets, which would aid in the development of long-term effective AD treatments. This includes AHR, a transcriptional factor exerting both anti-inflammatory and pro-inflammatory activities dependent on the ligand activating the receptor [95]. Being expressed in keratinocytes, it represents a worthwhile topical treatment approach for AD. A natural AHR agonist, tapinarof, has demonstrated a significant anti-inflammatory response in both pre-clinical and clinical studies, reducing the aggravation of disease symptoms when applied topically [96]. Clinical trial data suggesting an optimistic outcome for this compound indicate that the best results can be expected with a 1% cream formulation [97].

As discussed previously, the active role of alarmins derived from keratinocytes such as IL-33, TSLP, and IL-25 in the initiation of immune reactions implies that these alarmins could also be potential targets for AD. This was further supported by the convincing results obtained from clinical trials for the drug candidate tezepelumab, an antibody targeting TSLP [98,99]. It was found to cause remarkable improvement in acute allergic asthma patients [100].

The manifold biological roles of IL-33, another alarmin derived from keratinocytes with a prominent contribution to the early manifestations of AD, provide another interesting therapeutic strategy. A single dose of etokimab, an anti-IL33 antibody, gave convincing results in moderate to severe AD patients. This positive impact lasted for 140 days post-single dose application. Etokimab was found to inhibit the migration of neutrophils in vitro, either directly or via CXCR-1 [101]. However, the role of this antibody in later stages of AD requires further studies. Apart from this, five different biologics and a few other anti-IL33 antibodies are yet to be investigated for their role in AD.

Skin dysbiosis or injury induces the keratinocytes to release a number of pro-inflammatory cytokines, among which IL-1α is the first and foremost mediator of the inflammatory cascade and antigen presentation [102]. This indicates its potential role in the therapeutic management of AD. Bermekimab, an anti-IL-1α antibody, has been shown in studies to be safe and effective in the treatment of AD, with a significant reduction in itching and pruritis [103].

Colonization with S. aureus during disease pathogenesis induces inflammation mediated by IL-1R and IL-36R [104]. The latter, being an active participant in the innate immune system, is increasingly prevalent in the skin of AD and psoriasis patients. Spesolimab, the anti-IL-36R antibody, was found to be effective in pustular psoriasis, a rare form of the disease, which promoted further studies on the role of this drug in AD [105].

Several studies have also demonstrated the potential role of commonly available natural products in treating atopic dermatitis either by suppressing inflammatory reactions, inducing apoptosis, or preventing microbial infections [106]. Mangiferin, obtained from *Mangifera indica*, can ameliorate the skin lesions prominent in dermatitis by reducing the inflammatory biomarkers such as IL-6, TNF-α, and IL-1β [107]. It can also significantly inhibit macrophage biomarkers, such as CD68, that contribute to atopic dermatitis [107]. Curcumin-containing herbal extract cream, Herbavate, was found to be effective in 150 dermatitis patients. However, firm evidence is lacking to prove the beneficial effect of curcumin [108]. A micro sponge gel formulation comprising naringenin, another natural compound, was also found to reduce inflammatory infiltration and epidermal thickening [109,110]. Quercetin and tannic acid, on the other hand, apart from exerting their anti-allergic effect, are potent mast-cell inhibitors [111]. They suppress the polarization of Th2 cells, reducing TSLP and TARC levels as well as neo-angiogenesis [112]. Apigenin can treat both acute irritant contact dermatitis and acute allergic dermatitis by reducing TEWL, making the skin surface acidic and improving skin hydration [113].

In addition to these targeted approaches, emollients or moisturizers continue to be the only effective treatment for epidermal barrier dysfunction, controlling dryness, water loss, and inflammation [114,115]. However, a recent report criticizes this traditional therapeutic role of moisturizers as a prophylactic measure in newborn infants prone to AD [116,117].

## 7. Conclusions and Future Perspectives

This article illustrates the role of keratinocytes in AD both as a benevolent barrier safeguarding against the pathogenic triggers and as a hostile contributor inducing allergic reactions. Impairment of the innate immune system and epidermal barrier function along with a rise in cutaneous inflammation serve as the major key factors behind the pathogenesis of AD. This emphasizes the role of keratinocytes, the innate immune cells of our skin, in AD. The prevalent treatments are directed towards suppression of the inflammatory reaction and consolidation of the barrier defense. A few targeted approaches, as mentioned in the article, point out the therapeutic importance of keratinocytes. Further insight into certain pathways activated by keratinocytes and the consequent cytokines released would facilitate the development of targeted therapy. However, the complexity of the symptoms and biological processes involved has hindered the process so far.

More emphasis should be placed on preventative measures, particularly in the case of newborn infants. Various types of prebiotics and probiotics, including hydrolyzed formulations, should be explored along with measures to enhance the skin barrier in neonates. An elaborate understanding of the pathways activated at different stages of AD is important for the development of promising drugs, as drugs exhibiting limited efficacy in adults might be beneficial in the pediatric population. The development of pharmacogenetics with an understanding of the causative genes involved in the therapeutic variation will also help to develop effective interventions targeting different phenotypes of AD.

## Figures and Tables

**Figure 1 cells-11-01683-f001:**
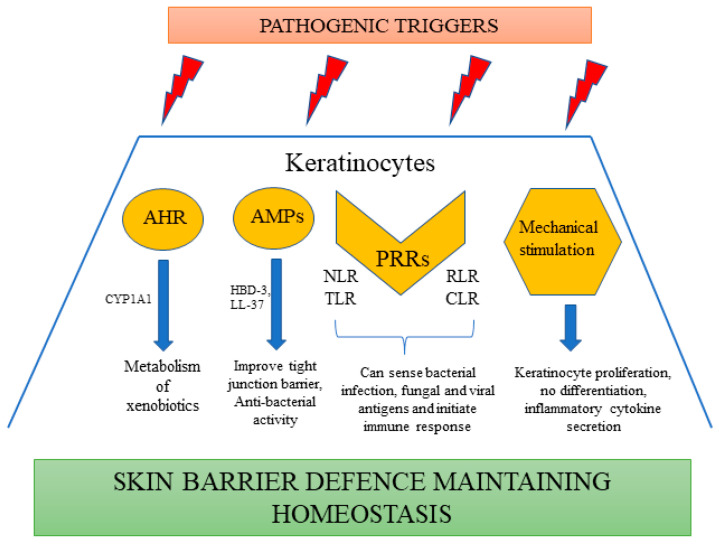
Keratinocytes as a part of skin immune defense.

**Figure 2 cells-11-01683-f002:**
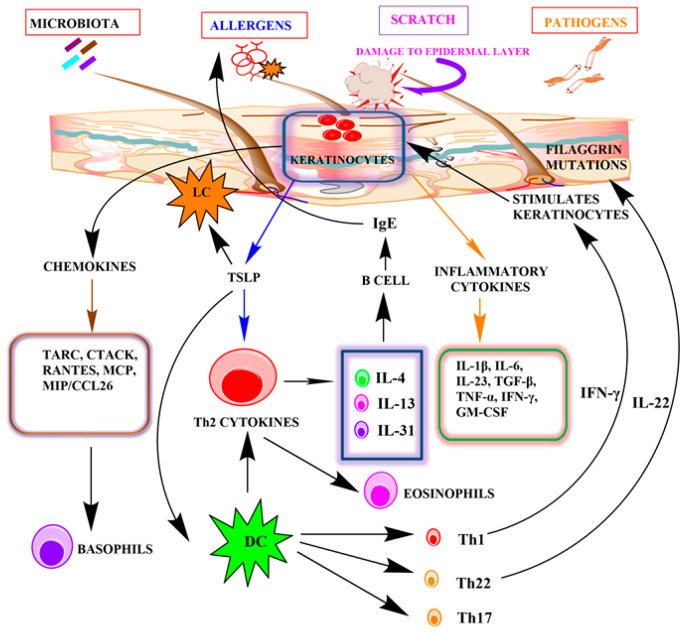
The role of keratinocytes in the pathogenesis of atopic dermatitis.

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
