# Peer review of "Keratinocytes: An Enigmatic Factor in Atopic Dermatitis"

_cells, 2022, doi:10.3390/cells11101683_

Round 1
Reviewer 1 Report
This article examines the role played by keratinocytes in the pathogenesis of atopic dermatitis to facilitate the opening of new therapeutic avenues. The introduction is clear as is the structure of the entire article. The text and the statements are supported by fairly recent references. Overall it is a good article.
I ask you to insert an additional paragraph to investigate any natural molecules that acting on cytokines and interlokines, which acting on keratinocytes or released by these cells, could influence the onset of atopic dermatitis.
Similarly, therefore, showing the influence of these molecules on these cells would still confirm the hypothesized role of keratinocytes in the pathogenesis of atopic dermatitis.
Reviewer 2 Report
Ms-cells 16584 can be accepted.
Author Response
Thank you for accepting our review article.